# Shedding Light on the Complex Regulation of FGF23

**DOI:** 10.3390/metabo12050401

**Published:** 2022-04-28

**Authors:** Marc G. Vervloet

**Affiliations:** 1Amsterdam UMC, Location Vrije Universiteit Amsterdam, Nephrology, De Boelelaan 1117, 1081 HV Amsterdam, The Netherlands; m.vervloet@amsterdamumc.nl; Tel.: +31-20-4442671; 2Amsterdam Cardiovascular Sciences, Diabetes and Metabolism, De Boelelaan 1117, 1081 HV Amsterdam, The Netherlands

**Keywords:** FGF23 (Fibroblast Growth Factor 23), regulation, mineral metabolism, PTH, DMP1, phosphate, calcium, vitamin D

## Abstract

Early research has suggested a rather straightforward relation between phosphate exposure, increased serum FGF23 (Fibroblast Growth Factor 23) concentrations and clinical endpoints. Unsurprisingly, however, subsequent studies have revealed a much more complex interplay between autocrine and paracrine factors locally in bone like PHEX and DMP1, concentrations of minerals in particular calcium and phosphate, calciprotein particles, and endocrine systems like parathyroid hormone PTH and the vitamin D system. In addition to these physiological regulators, an expanding list of disease states are shown to influence FGF23 levels, usually increasing it, and as such increase the burden of disease. While some of these physiological or pathological factors, like inflammatory cytokines, may partially confound the association of FGF23 and clinical endpoints, others are in the same causal path, are targetable and hence hold the promise of future treatment options to alleviate FGF23-driven toxicity, for instance in chronic kidney disease, the FGF23-associated disease with the highest prevalence by far. These factors will be reviewed here and their relative importance described, thereby possibly opening potential means for future therapeutic strategies.

## 1. Introduction

Fibroblast Growth Factor 23 (FGF23) has emerged as an important biomarker in chronic kidney disease (CKD) [1]. Accumulating evidence suggests that it not only is a risk predictor for cardiovascular disease, in particular heart disease and heart failure, but also a uremic toxin itself, directly causing disease [2]. For both properties, being either an independent risk predictor or a direct toxin, in-depth knowledge of its regulation is of paramount importance. In the setting of FGF23 as an independent risk factor, but not directly inflicting harm, its association with clinical endpoints is confounded by hitherto hidden mechanisms that are in the causal path to these endpoints. Exploring these regulators of FGF23 may thus reveal novel targets of treatment and hold the promise of improving outcomes for patients with kidney disease. In turn, if FGF23 itself is the causative molecule, intervening in its regulators may also modify FGF23-driven morbidity.

Besides being a prominent hormone in CKD, the discovery of FGF23 solved the quest for a humoral factor explaining several inheritable diseases characterized by renal wasting, which by then could be explained by mutations of FGF23 itself or factors involved in its regulation [3].

FGF23 is a hormone, secreted by osteocytes, and has a central physiological role in phosphate homeostasis. It promotes phosphaturia and inhibits the activation of vitamin D, thereby limiting vitamin-D mediated phosphate absorption from the diet by the transcellular uptake route of enterocytes in the gastro-intestinal tract. There are several principal ways in which FGF23 concentrations can be regulated, and all of these appear to play a role. These modes of regulation are production and secretion by the cells of origin, ectopic production, and cleavage or breakdown at cells of origin or after release into the circulation. The currently available immunoassays measure either the full-length and biologically active hormone, termed intact FGF23 (iFGF23), or both iFGF23 and its c-terminal fragment, termed cFGF23. It should be noted that the term cFGF23 for this assay is a confusing term, because it does not only measure the c-terminal fragment, the latter originating after cleavage of the full-length polypeptide. This cleavage obviously also generates an N-terminal fragment, but no commercially available assay detects this fraction. Although intact FGF23 is assumed to be the physiological effector molecule, debate exists on the role of the fragments, possibly having agonistic or antagonistic effects, the latter as a competitive inhibitor [4].

## 2. The Role of Minerals as Regulators of FGF23

### 2.1. Phosphate

Given the key role of FGF23 to protect the organism against hyperphosphatemia, it can be expected that phosphate increases FGF23 concentrations. Indeed, several studies have shown that an increase in dietary intake of phosphate by both healthy volunteers and people with CKD increased its concentrations, albeit with some delay of around 24 h [5,6,7] to restore phosphate balance. In turn, phosphate restriction has the ability to lower FGF23, but, different from PTH secretion from healthy parathyroid glands in a setting of hypercalcemia, FGF23 has never been described to be fully suppressed following hypophosphatemia, for instance when induced by mutations in the renal phosphate transporter NaPi2c which gives rise to hereditary hypophosphatemic rickets with hypercalciuria (HHRH) [8,9]. Until recently, the underlying molecular mechanism by which phosphate modulates FGF23 levels has been elusive. It has been shown that phosphate transport into bone cells across the inorganic phosphate transporter 1 (PiT-1) may be involved [10]. A recent study revealed an additional remarkable mechanism [11]. Bone cells that produce FGF23 express its receptor FGFR1 as well. It is now shown that phosphate itself can bind to this unliganded receptor, leading to the upregulation of the *Galnt3* gene, the protein-product of which leads to O-glycosylation of full-length FGF23, as will be discussed below. The consequence of this post-translational modification of the FGF23 molecule is that it escapes intracellular cleavage, increasing the proportion of biologically active FGF23. This mechanism does not suggest that phosphate induces FGF23 expression, even though a previous study suggested it can in a cell line [12], but rather stabilises the hormone. This mode of action of phosphate on FGF23 concentrations is in line with clinical studies in patients with CKD that addressed the question of whether dietary phosphate restriction can lower FGF23. A recent meta-analysis of these studies found more pronounced reduction of iFGF23 than of cFGF23, the latter also measuring FGF23 fragments [13]. In normophosphatemic CKD patients, short-term treatment with non-calcium containing phosphate binders did not change FGF23 [14,15], while prolonged treatment induced a substantial decline [16]. The use of calcium-containing binders did increase FGF23 [17].

### 2.2. Calcium

Interestingly, there appears to be a minimal concentration of calcium required for phosphate to be able to increase FGF23 levels. In an animal model testing varying serum concentration of calcium, it was shown that an increment of FGF23 by PTH was completely abolished when ionized calcium concentrations were below 4 mg/dL [18]. The physiological functionality of this phenomenon might be that this prevents the catabolism of vitamin D by FGF23 in a setting of hypocalcemia. Moreover, in an animal model, calcium itself was shown to be able to directly increase FGF23 transcription by acting on the promotor of the *Fgf23* gene [19,20]. These findings from experimental research are in line with most, but not all, clinical observations. In a clinical trial among 30 early CKD patients, studying the effects of adding calcium carbonate to calcitriol, it was shown that this induced an increase of FGF23, which was paralleled by an increase in serum calcium concentration [21]. In more advanced CKD, the non-calcium containing phosphate binder lanthanum carbonate was able to lower FGF23 levels, while a calcium-containing binder could not [17]. However, in a short-term study, acute increments or decrements of serum calcium concentrations had no effect on FGF23 [22].

### 2.3. Calciprotein Particles

Apart from the synergistic effects of combined higher levels of calcium and phosphate on increasing FGF23, it is possible that this is mediated by the formation of calciprotein particles (CPP) [23,24]. Even at physiological concentrations, human plasma is supersaturated for calcium and phosphate, which would induce spontaneous hydroxyapatite crystal formation [25]. These potentially damaging crystals, however, are prevented from being formed and freely floating in the circulation by being scavenged into soluble amorphous primary calciprotein particles CPP (CPP1), which are nanoparticles containing the serum protein Fetuin-A as the main protein constituent. In a setting of increased availability of these minerals, as is the case for phosphate in CKD, or suppressed hepatic production of Fetuin A in a setting of chronic inflammation, the stage is set to overwhelm the capacity of this defence system, leading to the formation of more toxic larger crystalline secondary CPPs (CPP2) [26,27,28]. Like high exposure to phosphate, exposure to high calcium levels also increases the amount of CPP, as was shown in a patient with renal sarcoidosis, and this was paralleled by an increase in FGF23 [29]. The role of calcium on the formation of CPP was also shown in a clinical study comparing calciumcarbonate with lanthanumcarbonate [30]. After switching to lanthanumcarbonate, the total amount of CPP declined substantially, without major differences in serum concentration of calcium and phosphate between the two phosphate binders.

A recent clinical observational study demonstrated an association between the amount of CPPs and FGF23, suggesting an induction in the phosphaturic hormone by CPP’s [31]. Indeed, a recent in vitro study found that CPPs are capable of increasing FGF23 expression in osteoblast-like cells [32]. Remarkably, this effect was restricted to the smaller sized CPP1. It is therefore conceivable that an increased amount of CPP’s formed triggers FGF23, which in turn induces phosphaturia and declines levels of active vitamin D. FGF23 thereby slows the formation of CPP’s by lowering the concentrations of the minerals that form its mineral content. This concept is supported by the ability of CPP to exit the circulation, enter the bone marrow and reach FGF23-producing bone cells [32].

### 2.4. Magnesium

Given its resemblance to calcium as a bivalent cation, and its beneficial effects on the formation of CPP [33,34,35], it is likely that magnesium is also involved in the regulation of FGF23. Data, however, are scarce. In an animal study of cats with chronic kidney disease, a negative association between serum magnesium concentration and FGF23 was found, which was independent of calcium, phosphate, and PTH [36]. In an observational study among young healthy men, it was shown that a lower dietary intake of magnesium was associated with higher FGF23 [37]. When rats were exposed to a short-term (7 day) magnesium deficient diet, FGF23 levels were higher compared to a normal diet at all time points following the interventions which reached statistical significance after one week [38]. However, clinical trials demonstrating a beneficial effect on clinically relevant endpoints of magnesium supplements are lacking [39].

The role of minerals and calciprotein particles are summarized in Figure 1.

## 3. Hormonal Regulation of FGF23

### 3.1. Parathyroid Hormone

PTH was shown to be a relevant regulator of FGF23 by directly increasing its expression in bone in an experimental model of CKD [40]. Moreover, in that same study, parathyroidectomy before the onset of CKD completely abolished the FGF23 increment, even in a subsequent setting of hyperphosphatemia. This is probably mediated by the receptor PTH1R for PTH on bone cells, the same receptor that is involved in regulating bone turnover, and with Nuclear Receptor Related-1 protein (Nurr1) an intermediate intracellular molecule [41]. Another established action of PTH on bone cells is the suppression of the gene encoding for sclerostin (*Sost*). Sclerostin acts as local inhibitor of the Wnt pathway by sclerostin, thereby suppressing FGF23 [42,43,44]. PTH therefore unleashes FGF23 by suppressing sclerostin. Clinical studies suggest a biphasic response to PTH. In a short term (3 h) 1–34 PTH infusion in healthy young persons, FGF23 declined, most likely driven by PTH-induced renal phosphate loss [45]. During this period 1,25 dihydroxyvitamin D3 (1,25D) started to rise, which expectedly would induce increased dietary phosphate uptake. This, and the potential direct effects of PTH on bone cells may be the dominating effect following more prolonged exposure, giving rise to FGF23 increments. This indeed was suggested by a two days PTH infusion study that led to increased cFGF23 in healthy persons and people treated by dialysis regardless of bone turnover status [46]. Like for many other aspects, however, the role of PTH is complex, because if endogenous levels rise as a consequence of a decline of serum calcium by sodium citrate infusion, FGF23 did not increase [22]. Obviously, the stimulating effects of PTH on FGF23 may have been nullified by the low levels of calcium. There seems to be a logical physiological basis for the induction of FGF23 by PTH. The key purpose of PTH is to restore hypocalcemia and it does so in part by liberating calcium form bone. This is paralleled by release of phosphate, which is, besides by phosphaturic effects of PTH itself, excreted by the kidneys under the influence of FGF23.

Observations of persons with dialysis-dependent end-stage kidney disease treated with calcimimetics appear to be in line with the notion that lowering PTH is accompanied by declining FGF23 [47,48]. Remarkably, however, in both of these clinical studies, using the oral cinacalcet or the intravenous etelcalcetide, the decline of FGF23 followed reductions of phosphate and calcium, instead of PTH reductions.

### 3.2. Vitamin D

There is strong evidence that 1,25D directly induces *Fgf23* gene transcription. Mice injected with the active form of vitamin D had increased levels of FGF23 mRNA, exclusively in bone, which was accompanied by a rise in serum FGF23 levels [49]. In that same study, rat-derived UMR-106 osteoblast-like cells had a 1000-fold increase of FGF23 mRNA 4 h after exposure to 1,25D. In another study with a focus on exploring the *Fgf23* gene promotor region, this role of 1,25D was confirmed [50]. Collins and co-workers observed three patients that received a high dose of calcitriol after parathyroidectomy after surgery and observed steep increments of FGF23 [51]. Many clinical trials have been performed in which either active or nutritional vitamin D was the key intervention. In several of these trials, FGF23 levels were part of the follow-up parameters. The results of these observations have been summarized in two meta-analyses. In the first of these it was found that in patients that were deficient in vitamin D at baseline, the intervention induced a statistically significant increase of iFGF23 [52]. There was also an increase of cFGF23, but this did not reach statistical significance. A very recent meta-analysis could not confirm this effect of vitamin D, but in this meta-analysis, trials were included where participants did not have vitamin D deficiency at baseline, which may explain the discrepancy with the previous analysis [53]. In a study among children treated by dialysis, active vitamin D compounds (calcitriol or doxercalciferol) induced a substantial increase in FGF23 [54]. Collectively, these studies strongly suggest that vitamin D, especially active vitamin D, induces FGF23.

A summary of the roles of PTH and vitamin D is provided in Figure 2.

## 4. Local Regulators of FGF23 in Bone

### 4.1. Factors Involved in FGF23 Expression

Dentin Matrix Protein 1 (DMP1) and PHosphate regulating gene with homologies to Endopeptidases on the X chromosome (PHEX) both are suppressors of FGF23 gene expression that appear to act in concert locally in bone for that function [55,56,57]. PHEX is also believed to promote FGF23 cleavage, which then would induce a lower iFGF23 over cFGF23 ratio. Mutations in either PHEX (XLH, X-linked hypophosphatemic rickets) and DMP1 (ARHR, autosomal recessive hypophosphatemic rickets) cause renal phosphate wasting and its clinical sequelae by primary elevations of FGF23. There are no descriptions in the literature of acquired malfunction or suppression of the PHEX protein, with the possible exception of a report on a patient with leprosy [58]. For DMP1, however, diseases that induce acquired suppression appear to exist. In a mice model of CKD, it was shown that renal failure lowered osteocyte DMP1 expression, followed by FGF23 increases, while supplementation of DMP1 partially restored FGF23 towards the normal lower range [59]. The extent to which this is of relevance in clinical CKD remains to be established, but it has been shown that lower circulating levels of DMP1 are associated with cardiovascular event [60], and this finding may be mediated by increases of FGF23. In addition, uremia induced suppression of DMP1, and hence increments of FGF23 may explain the clinical observation that in more early stages of CKD, FGF23 and phosphate levels appear to diverge, pointing to another inducer of FGF23 than phosphate itself, namely suppressed DMP1 [61]. *α*-Klotho is intricately involved in phosphate homeostasis and the biological activity of FGF23 [1]. Its colocalization with FGFR1 is mandatory for signal transduction of FGF23 across the cell membrane to exert its actions in the proximal tubules, to induce phosphaturia. Recent research has now revealed that the circulating form of *α*-klotho, generated after cleavage of its large ectodomain [62], is involved in the expression and excretion of FGF23 from osteocytes. This hitherto unknown role of *α*-klotho was postulated after analysis of a 13-months old girl with unexplained elevation of FGF23 leading to hypophosphatemic rickets [63]. She was found to have a translocation nearby the *α*-*klotho* gene. This phenotype could be mimicked in an animal model by using an adenovector-induced increased expression of *α*-klotho, leading to high levels of circulating *α*-klotho, accompanied by a very steep rise of FGF23 and hypophosphatemia [64]. A recent study employed targeted deletion of the *α*-*klotho* gene from long bones and found that this led to attenuated increase of FGF23 after induction of CKD, both at osteocyte expression level and its circulating concentration [65]. This strongly suggests that *α*-klotho is required in an autocrine fashion for FGF23 expression from osteocytes. Both studies revealed that the presence of FGFR1 on osteocytes is required.

### 4.2. Post-Translational Modification of FGF23 in Bone

Following the translation of FGF23, the full-length polypeptide can be cleaved intracellularly before being secreted, thereby preventing the biologically active compound to enter the circulation. This cleavage occurs between the arginine residues at positions 176 and 179, and mutations at either of the arginine residues renders FGF23 resistant to proteolytic cleavage, giving rise to autosomal dominant hypophosphatemic rickets (ADHR) [66]. This cleavage is assumed to occur at the Golgi-apparatus by one of seven serine-proteases belonging to the family of subtilisin-like preprotein convertases (SPC), which act by cleaving polypeptides from preproteins to its mature polypeptide backbone. The most likely SPC is furin because its knock-out completely prevented FGF23 cleavage [67]. Prior to being exposed to these proteases, in particular furin, FGF23 can be O-glycosylated by N-acetylgalactosaminyltransferase 3 (GalNT3) at threonine residue position 178, which induces resistance to proteolytic cleavage of FGF23. As indicated above, exposure to phosphate may increase this O-glycosylation and thereby increase the relative amount of full-length FGF23, the active form, as a feedback mechanism to restore phosphate to lower concentrations. In turn, FGF23 can also be phosphorylated at a serine residue at position180 by a kinase termed Fam20c, which prohibits O-glycosylation by GalNT3, which ultimately makes FGF23 more prone for proteolytic cleavage [67].

## 5. Clinical Conditions and Their Impact of FGF23

### 5.1. Anemia and Iron Deficiency

Patients with ADHR, one of the inherited forms of renal phosphate wasting due to inappropriate elevations of FGF23, can present rather late (from puberty or not even before their mid-forties), and frequently do not present with typical features such as short stature or bowed deformations of the lower extremity [68]. While these patients have limited or absent capacity to cleave FGF23, it is assumed that as long as the baseline transcription of FGF23 is rather low, circulating iFGF23 can remain relatively normal for years without severe phosphate losses. Iron deficiency in these patients was associated with increased iFGF23 levels [69] and in a small open label trial oral iron supplementation substantially lowered FGF23 level in patients with ADHR [70]. These clinical observations are in line with animal research on models of ADHR [71]. In that experimental study it was additionally shown that exposure of osteoblastic cells (UMR-106) to low iron condition increased mRNA of FGF23 up to 20-fold. The mechanisms involved were mitogen-activated protein kinase (MAPK) dependent. In addition, iron-deficiency also induced increments of Hypoxia Inducible Factor 1*α* (HIF1*α*), and HIF1*α* itself could also boost FGF23 expression. Indeed, it was shown that a HIF1*α* binding site exists in the promotor region of the *Fgf23* gene [72]. In addition HIF1*α* prevents the cleavage of FGF23 [73]. Collectively, these findings would lead to higher circulation levels of iFGF23 in a setting of increased expression of HIF1*α* by either iron deficiency or hypoxia. However, in a study using the HIF1*α* stabiliser molidustat in an animal model of CKD and in additional in vitro experiments, it was shown that improved iron availability to osteocytes by the compound abolished the increased FGF23 expression [74]. This same study also revealed that EPO increased FGF23 [74], and this finding was previously shown in both patients and animal models [75]. This latter study demonstrated that this effect was sustained after bone marrow ablation, where upregulation of the *Fgf23* gene persisted, strongly suggesting a direct effect on these cells in cortical bone. Also in human studies, either EPO levels or exogenous doses were associated with FGF23, in particular total FGF23, while the effects on iFGF23 were indeterminant [76]. The role of hepcidin, a liver-derived acute phase protein that induces functional iron deficiency, as an intermediate metabolite in anemia, and iron-deficiency associated FGF23 upregulation is not yet well established.

### 5.2. Inflammation

Several reports point to the role of inflammatory mediators on bone cells leading to increased expression and secretion of FGF23 [73,77]. In turn, FGF23 can upregulate inflammatory mediators from hepatocytes [77]. Especially in the setting of advanced CKD with remarkably high concentrations of FGF23, this may initiate a pro-inflammatory vicious circle, further driving FGF23. Indeed, several pro-inflammatory cytokines such as tumor necrosis factor (TNF), Interleukin-1β (IL-1β), TNF-like weak inducers of apoptosis (TWEAK) and also bacterial lipopolysaccahrides (LPS) have been shown to stimulate both *Fgf23* gene expression and protein excretion in a cell model of osteocytes [78]. In another study, LPS injections increased FGF23 despite a low phosphate diet [79]. Interestingly, the exposure to LPS also caused renal FGF23 resistance by suppression of kidney *α*-klotho, thereby dismantling the FGF23 receptor.

Using several animal models of CKD, or TNF injections in mice with normal kidney function, it was found that TNF increased FGF23 while anti-TNF prevented this [80]. Importantly, the source of FGF23 in that study was the kidney itself, possibly driven by the highest local concentrations of TNF in that organ. This role of TNF is in line with the identification of a TNF responsive FGF23 enhancer, suggesting the direct upregulation of FGF23 by this inflammatory cytokine [81], although it has also been suggested that increases of NF-κB are required.

### 5.3. Chronic Kidney Disease

An extensive review of the impact of chronic kidney disease on FGF23 is beyond the scope of this review and has been extensively reviewed recently [1,82]. Besides the propensity to accumulate phosphate as a driver for FGF23 increases, in addition to hyperparathyroidism, DMP1 suppression, as outlined above, chronic inflammation, iron deficiency, and FGF23 resistance due to *α*-klotho deficiency have all been implicated in the exponential rise of FGF23 as CKD progresses. Importantly, experimental studies found that FGF23 cleavage in CKD is impaired as it is in ADHR [73,83]. This feature of CKD, the precise molecular mechanisms of which is currently unknown, fits with the observation that in end stage kidney disease, most circulating FGF23 is intact [84]. Interestingly, it was recently shown that in a model of acute kidney injury, the kidneys themselves produce glycerol-3-phosphate (G3P), which directly stimulates FGF23 production, exclusively in bone [85]. It is likely that besides novel regulators like G3P, the impact of CKD on many, if not all, of the mechanisms involved, as described above, is huge, and collectively creates a perfect storm for essentially unopposed upregulation of FGF23. In addition, it seems plausible that in the setting of CKD, the cleavage of FGF23 is attenuated or its capacity overwhelmed, leading to extremely high levels of biologically active FGF23 in end stage kidney disease, most likely contributing to uremic toxicity.

## 6. Conclusions

The physiological regulation of bone-derived FGF23 is complex, and is regulated at levels of gene transcription, post-translational modifications, cleavage and cellular release. In addition, remote biological activity is variable by dynamic affinity of its receptor due to changing *α*-klotho abundance, possibly competitive inhibition by FGF23-fragments, and also varying expression of the FGF receptors themselves [86]. Moreover, ectopic FGF23 production has been described too, as outlined for the kidney as described above, but cardiac production has also been described [87,88]. The machinery involved in regulating the metabolism of FGF23 involves an intricate interplay between minerals, calciprotein particles, the endocrine system and local regulators in the vicinity of osteoblasts and osteocytes in an autocrine or paracrine fashion. Since FGF23 is most likely involved in the pathogenesis of an expanding list of diseases, in-depth knowledge of these regulatory pathways is the first step in ultimately targeting these molecular mechanisms that are in the path to clinical events. The exploration of these pathways is far from being finalized, and designing safe and effective interventions are only at the beginning.

## Figures and Tables

**Figure 1 metabolites-12-00401-f001:**
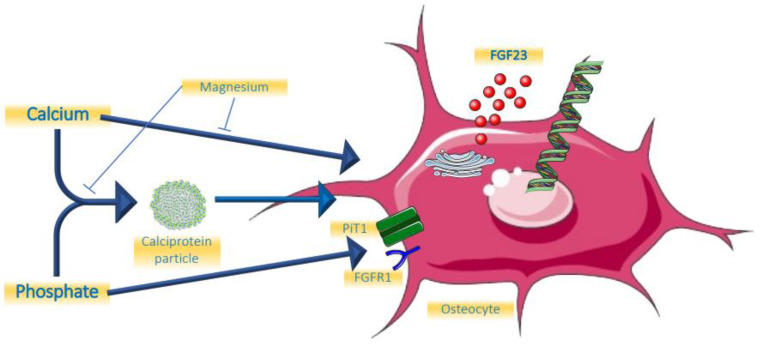
Effects of minerals on FGF23.

**Figure 2 metabolites-12-00401-f002:**
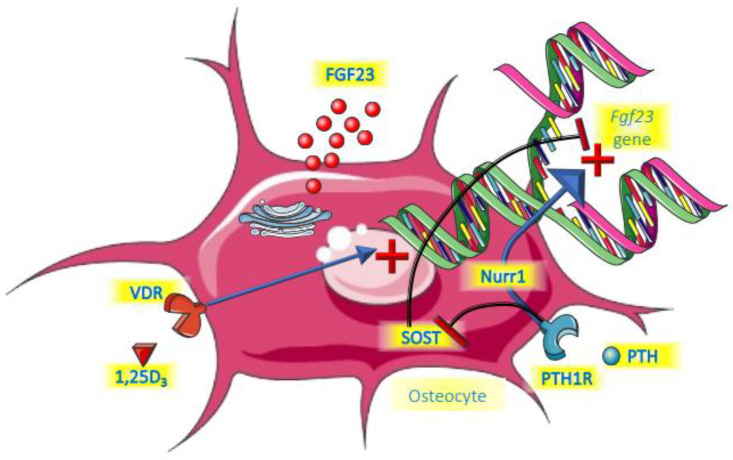
Endocrine control of FGF23.

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
