# Peer review of "Shedding Light on the Complex Regulation of FGF23"

_metabolites, 2022, doi:10.3390/metabo12050401_

Round 1
Reviewer 1 Report
This manuscript is well written and informative for clinician and nephrologist. FGF23 is reported to be associated with LVH and vascular calcification. In addition, recent study showed the role of FGF23 for infection and fracture.
page1 line38 renal wasting wasting→renal wasting page2 line62 [parathyroid glans]→parathyroid glands page2 line92, page5 line190, page7 line276, page7 line297 FGF23?「Fgf23」 7 line293 [FGF3]→FGF23
Author Response
Many thanks to reviewer 1: See the point by point reply attached

Reviewer 2 Report
The paper I was asked to review deals with the complex regulation of FGF23 production and secretion. It was interesting to read and quite clear despite providing a reader with a lot of detailed and complex knowledge. I think it is worth publishing.
some details to be corrected
line 37 "wasting wasting"
line 117 "ins"
line 223 "oof"
line 236 "rised"
line 326 - would gladly see some a citation at the end
line
Author Response
Many thanks to the reviewer, see the attached reply to the issues raised

Reviewer 3 Report
Dear editors:
It is a great honor and pleasure for me to be invited as the reviewer for this important manuscript entitled “Shedding light on the complex regulation of FGF23”. Marc Vervloet comprehensively review the mineral-FGF23-Vitamin D-PTH axis for chronic kidney diseases-mineral bone disease (CKD-MBD) and gain a deeper mechanistic insight into future potential therapeutic targets. This study topic is interesting and novel, attributing to Prof. Vervloet’s long-term efforts and contributions in this scientific field. Although the rationale is well-written, I have a number of comments concerning this study:
- In light of the scope of this study, the title of the article could be rephrase as “Shedding light on the complex regulation of FGF23 in Chronic Kidney Disease-Mineral Bone Disease” to intensify the extent to which the research area will be explored in the work. CKD-MBD represents a systemic disorder of mineral and bone metabolism due to CKD manifested by a myriad of complications as follows:
- Abnormalities of calcium, phosphorus (phosphate), parathyroid hormone, or vitamin D metabolism
- Abnormalities in bone turnover, mineralization, volume, linear growth, or strength
- Vascular or other soft-tissue calcification
- There too many grammar, spelling and syntax errors throughout the manuscript, e.g., “FGF23-assocted” disease in Line 19, “Unsurprisingly however” in Line 10…..
- Line 26: it not only is a risk predictor for cardiovascular disease, in particular heart disease and heart failure, but also a “uremic toxin itself, directly causing disease”. The author should cite an important reference, which proves that beyond mineral dysregulation, uremic vascular calcification can be solely the stimulation of PCS, reflecting PCS is a pro-calcific toxin: ” Chang JF, Hsieh CY, Liou JC, Liu SH, Hung CF, Lu KC, Lin CC, Wu CC, Ka SM, Wen LL, Wu MS, Zheng CM, Ko WC. Scavenging Intracellular ROS Attenuates p-Cresyl Sulfate-Triggered Osteogenesis through MAPK Signaling Pathway and NF-κB Activation in Human Arterial Smooth Muscle Cells. Toxins (Basel). 2020 Jul 24;12(8):472. doi: 10.3390/toxins12080472. PMID: 32722241; PMCID: PMC7472002. Th reference also concludes PCS induces osteogenesis through triggering intracellular intracellular ROS, pERK MAPK pathways and nuclear translocation of NF-κB to enhance downstream Runx2 and ALP expressions, which corresponds the description in Line 308: it has also been suggested that increases of NF-κB are required.
- Line 128: Although a negative association between serum magnesium concentration and FGF23 was found, magnesium therapy had side effects and a risk of hypermagnesemia in CKD population. It is not a standard treatment for CKD-MBD without evidence-based medicine. According to KDIGO guidelines, phosphate-binder therapy to lower calcium-phosphate products is the first choice. Due to impracticality and controversy, the inhibitory effect of magnesium supplement should be expressed as “dash line” in Fig 1.
- As mentioned above, the author must highlight the importance of novel phosphate-lowering therapy in accordance of the current guidelines.
- After minor revision and English editing, the article might be considered for publication.
- I found that "Keywords" were absent behind the section of the abstract, please add it.
Author Response
Many thanks to the several useful suggestions, which I responded to in the attached report
